**Data Availability Statement:** Data are available from the KNH/UON ethical review committee (contact via uonknh_erc@uonbi.ac.ke) for

# Adolescent perspectives on peripartum mental health prevention and promotion from Kenya: Findings from a design thinking approach

Joseph Kathono[1], Vincent Nyongesa[2], Shillah Mwaniga[1], Georgina Obonyo[3], Obadia Yator[2], Maryann Wambugu[3], Joy Banerjee[4], Erica Breuer[5], Malia Duffy🔾[6,7], Joanna Lai[8], Marcy Levy[8], Simon Njuguna[9], Manasi Kumar🔾[2,10]*

1 Nairobi Metropolitan Services, Nairobi, Kenya, 2 Department of Psychiatry, University of Nairobi, Nairobi, Kenya, 3 Our Voices Initiative, Nairobi, Kenya, 4 Woodapple (Consulting), New Delhi, India, 5 University of Newcastle, Newcastle, New South Wales, Australia, 6 St Ambrose University, Davenport, Iowa, United States of America, 7 Health Across Humanity, LLC, Boston, Massachusetts, United States of America, 8 UNICEF Headquarters, New York, NY, United States of America, 9 Division of Mental Health, Ministry of Health, Nairobi, Kenya, 10 Institute for Excellence in Health Equity, New York University Grossman School of Medicine, New York, NY, United States of America

* Manasi.Kumar@nyulangone.org

## Abstract

In Kenya, approximately one in five girls aged 15–19 years old are pregnant or already a mother. Adolescent girls and young women experience significant mental health vulnerabilities during the pregnancy and postpartum periods, leading to poor antenatal and postnatal care attendance and inferior infant and maternal health outcomes. Pregnant adolescents often experience stigma and disenfranchisement due to their pregnancy status and at the same time lack access to mental health support within health settings, schools, religious institutions, and communities. This paper presents the results of qualitative interviews embedded within the human-centered design (HCD) process used to adapt the Helping Adolescents Thrive (HAT) program for Kenyan peripartum adolescents including young fathers. This qualitative study used two phases. First, a HAT advisory group participated in a series of four workshops to help identify and articulate mental health promotion needs and deepened the team's understanding of youth-centered thinking. Second, qualitative interviews were conducted with 39 pregnant and parenting adolescents to understand their perspectives on mental health prevention and promotion. Pregnant and parenting adolescents articulated different needs including poor support, stigma, and psychological disturbances. Parenting adolescents reported disturbed relationships, managing motherhood, poor health, and social empowerment. Participants highlighted sources of stress including economic challenges, fear of delivery, strained relationships, rejection, and stigma. Participants described psychological disturbances such as feeling stressed, worthless, withdrawn, and suicidal. Coping mechanisms reported by participants included engaging in domestic activities, hobbies, and social networking. Peers, family and spirituality were identified as important sources of support, as well as school integration, livelihoods, support groups and mentorships. Findings from this study can be used to strengthen and adapt HAT program,

researchers who meet the criteria for access to confidential data. Data are also available from Dr. Obadia Yator (contact via obadiayator@gmail. com).

**Funding:** The qualitative inquiry is part of an embedded study funded led by UNICEF and the inquiry reported here was partially supported by UNICEF Innocenti office. Research reported in this publication was supported by the Fogarty International Center of the National Institutes of Health under Award Number K43TW010716, which also supported the contributions of MK to this work. The content is solely the responsibility of the authors and does not necessarily represent the official views of the National Institutes of Health. The funders had no role in study design, data collection and analysis, decision to publish, or preparation of the manuscript.

**Competing interests:** The authors have declared that no competing interests exist.

policy and practice for mental health prevention and promotion for pregnant and parenting adolescents.

---

# 1. Introduction

Approximately 75% of mental disorders, including anxiety and depression, emerge before the mid-20s and two-thirds of adolescents do not receive mental health support when it is needed [1, 2]. Adolescent girls experience significant vulnerabilities in their mental health during the pregnancy and postpartum periods, which can lead to poor antenatal and postnatal care attendance, and poor infant and maternal health outcomes including suicide, which is the fourth leading cause of death among adolescents ages 15–19 globally [3–5]. Global prevalence studies demonstrate a wide variation in estimates of postpartum depression among adolescents, ranging between 14–53%. It has been found that adolescent girls are up to twice as likely to experience postpartum depression in comparison to adult postpartum women [5].

Mental health promotion activities can play a critical role in preventing poor health outcomes for both the mother and the infant however, there is limited guidance to address the unique mental health vulnerabilities of pregnant and parenting adolescent girls and young women. This gap is even more evident in low-income country settings where approximately 16 million adolescent girls and young women ages 15–19 give birth each year, experiencing confounding stressors, such as high poverty rates, lack of available resources, forced marriage, and service inaccessibility, among others that contribute to poor mental health [6].

In Kenya, approximately 20% of adolescent girls ages 15–19 are pregnant or are already mothers [7]. Pregnant adolescents in Kenya often experience stigma due to their pregnancy status contributing to disenfranchisement within institutional, social, and policy frameworks. Such neglect leads to a dearth of mental health interventions within health settings, schools, religious institutions, and communities [8]. Among adolescents and adults, integration of mental health screening and services within antenatal and postnatal care is feasible, acceptable, improves depression, and increases uptake of health services [9, 10]. Another study highlights the potential for social support interventions to advance mental health prevention and promotion during pregnancy and postpartum [11]. A recent program evaluation in Kenya on a home visiting team approach for pregnant and postpartum adolescent girls and young women integrated multiple services including mental health support and referrals as part of the case management activities targeting the adolescent, their partner, and family, highlighting a potential intervention to strengthen the adolescent's social ecology [12]. In the context of Kenya however, further understanding factors from the perspective of pregnant and parenting adolescents themselves that help prevent mental stress and promote mental health are critical to optimize antenatal and postnatal care attendance so that it supports the mental health of pregnant and postpartum adolescent girls and improves and infant and maternal health outcomes [12–14].

## 1.1 Objectives of the study

This study took place within the contexts of two related studies: 1) the Helping Adolescents Thrive (HAT) Kenya program (jointly delivered by the World Health Organization and UNICEF); and 2) the INSPIRE study ('Implementing mental health interventions for pregnant adolescents in primary care LMIC settings'). The focus of both studies is on peripartum adolescent girls and young women in Kenya. INSPIRE focuses on developing treatment

interventions for this cohort. Helping Adolescents Thrive (HAT) is meant to provide strategies, guidelines and tools to promote and protect adolescent mental health and reduce self-harm and other risk behaviors [15]. WHO and UNICEF developed guidelines called HAT that focus on a framework of action, implementation strategies as well as recommended actions across sectors to improve mental health outcomes for adolescents through preventive and promotive activities. The strategies include a) implementation and enforcement of policies and laws suggesting a whole-of-government and whole-of-society approach; b) improvements in the quality of environments in schools, communities and digital spaces; c) strengthening caregiver support, knowledge, competency and relationship with adolescents; and d) development of evidence-based psychosocial interventions for universal, targeted and indicated promotion and mental health prevention [15]. We used the HAT framework to examine the perspectives of pregnant and parenting adolescents through a human-centered design process to achieve two objectives: a) to gather information from pregnant and parenting adolescent girls and young women on the key contributors to mental stress and coping strategies that support their mental wellbeing; and b) to identify preferences for mental health prevention and promotion activities within HAT that are responsive to the challenges, and expressed needs of pregnant and parenting adolescent girls and young women as well as adolescent boys and young men.

The information gathered can help to inform essential mental health policy, program, and practice priorities to address the mental health promotion needs of pregnant and parenting adolescents in Kenya.

## 2. Method

### 2.1. Settings

The workshops and key informant interviews were conducted among adolescents from Kariobangi and Kangemi health centers. Allocated space within each health center was used to ensure visual and auditory privacy. Kariobangi health center is a level 3 facility under the Nairobi Metropolitan Services. It is located in a low-income residential area in northeastern part of Nairobi, Kenya. The area includes both lower-middle class and slums. Kariobangi north has a population of 18,903 residents [16]. Kangemi health center is also a level 3 facility under the Nairobi Metropolitan Services. The center is located in a slum in Nairobi City, within a small valley on the outskirts of the city. Kangemi area has a population of 116,710 residents [16]. The work leveraged ongoing efforts towards integrating perinatal adolescent mental health in these two clinics that is part of a Fogarty funded study that this team is working on [17].

### 2.2. Study design

This qualitative study employed a human-centered design, which uses a non–linear, emergent thinking process that can help researchers explore deeply, continually empathize, rapidly ideate, simply prototype, and constantly iterate [18]. For ease of communication and networking with adolescent participants we formed a WhatsApp-based group to complement workshops. Table 1 describes the two phases of research.

Fig 1 Describes the design process.

Policy maker interviews aimed to determine if the solutions identified by the workshop and qualitative interview participants were supported by national policies. Findings from qualitative interviews with policy makers are documented in a separate manuscript [19].

**Table 1. Research phases and associated activities.**

| Phase | Key Activities | Activity Description |
|---|---|---|
| 1 | Formed an advisory board comprised of pregnant and parenting adolescents, male partners/ adolescent fathers, caregivers, youth advocates, health workers, and a select group of researchers from the HAT team. | Met 4 times over a 6-month period. The adolescent workshops used focus group discussions (FGDs), role-plays, and journals of adolescents' daily activities. They helped define the problem we were aiming to find a design solution for, explore the current situation, and identify what needs to be changed or fixed. |
| 2 | The study team conducted key informant interviews (KII). | KIIs were selected from a separate sample of pregnant and postpartum adolescents in clinical settings to sharpen our understanding of individual mental health promotion and support needs of peripartum adolescents. |

## 2.3. Participant recruitment

**Phase 1: Adolescent advisory workshops and design thinking meetings.** Youth advocates recruited members they had worked with previously in sexual and reproductive health and rights (SRHR) programming for the adolescent advisory board, also adolescent girls whom we had interfaced with from settlements where the INSPIRE study on adolescent pregnancy is ongoing. Individuals were eligible for the advisory board if they were between 13–19 years and pregnant or within 12 months postpartum, fathers of the babies, or caregivers of pregnant or parenting adolescents. Participants had to consent to participate and be willing to share their thoughts and recommendations with the study team.

**Phase 2: Key informant interviews.** Respondents for qualitative interviews were recruited through community health assistants and community health volunteers working with Kangemi and Kariobangi health centers who identified eligible participants from the community, the antenatal clinic, and mother and child health clinic. Participants were eligible if they were between 13 and 19 years of age, pregnant or within 12 months postpartum, and consented to participate. The eligible participants were taken through consenting process by the interviewers in the health facilities. In Kenya pregnant adolescents who are below 18 years are considered emancipated minors and are legally able to consent on their own. We offered transport reimbursement and refreshments to the participants upon completion of each interview. Participant reimbursement is in line with guidelines of Kenyatta National Hospital/University of Nairobi ethical review committee which approved this study.

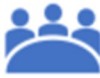 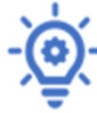 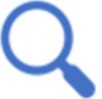 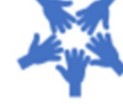

Theory of change (larger framework and stakeholder participation)

Get into design thinking process through identifying problem and range of solutions (design thinking process)

Design phase 1 – identify the problem and a clear problem statement

(adolescent and research team led workshops)

Design phase 2 – clear articulation of needs and problems with possible solution entry points

(co-creation of needs and challenges list)

**Fig 1. Human centered design workshop and design process.**

## 2.4. Workshops and key informant interview frameworks

**Phase 1: Adolescent advisory workshops and design thinking meetings.** The first workshop with adolescents, mental health specialists, and policymakers was held in April 2021 (Table 2). The consecutive workshops consisted of adolescents and caregivers, and took place in May 2021, the third workshop held over three days in August 2021, and the fourth workshop in September 2021. We provided travel stipends, refreshments, and infant care during the workshops. To optimize safety during the COVID-19 pandemic, we also provided hygiene products such as hand sanitizers and facial masks, in addition to social distancing measures with no more than 20 people in one room with some participants joining virtually.

**Phase 2: Key informant interviews.** The key informant interview framework focused on challenges and solutions to mental health during the pregnancy and postpartum periods. Adolescents were asked how they would like their life to be, including interpersonal relationships with their parents, partner and child, and ideas around how they could optimize their mental wellbeing and life opportunities. The interviews were carried out in the months of September and October 2021 within the two health care facilities. See S1 Table for the full key informant interview guide.

## 2.5. Data collection and analysis

**Phase 1: Adolescent advisory workshops and design thinking meetings.** Given the iterative nature of design thinking processes, an initial problem statement was defined with further refinements made throughout subsequent workshops. This started with the development of a draft Theory of Change for the intervention which will be described in another manuscript. Focus group discussions (FGDs) were interspersed with group meetings and brainstorming sessions to define the final problem statement and recommended solutions in accordance with design thinking methodology (see Fig 2 on brainstorming activities). Emphasis was placed on demonstrating the evolution of thought and journey for defining final solutions. Researcher took notes during the interactive and participatory brainstorming workshops and observation of role-plays. Adolescent workshop participants also submitted their journals to the researchers. These journals described their daily activities over a period of twenty-eight days to help illuminate their thoughts and emotions. Data from flipcharts and recorded workshop sessions were also converted into transcribed notes where data was de-identified. Information captured through Miro during virtual meetings further reflect adolescent articulation of needs, problems and consensus building around prominent mental health problems and priority.

**Table 2. Workshop agenda and participant description.**

| Workshop | Agenda | Participants |
|---|---|---|
| Workshop1: April 2021 | Develop a preliminary Theory of Change to review understanding, evidence and stakeholder perspectives on mental health promotion and insights to inform key components of a HAT promotion model | (n = 28) pregnant adolescents (n = 4), adolescent mothers (n = 4), male partners (n = 2), caregivers (n = 2), community stakeholders (n = 1), Ministry of health (n = 4), Nairobi Metropolitan services (n = 3), UN (n = 2), Community based organizations (CBOs) and mental health advocacy stakeholders (n = 4), theory of change consultant (n = 1), design thinking consultant (n = 1) |
| Workshop2: May 2021 | Exploring additional problems and review of Theory of Change map, introduce design thinking model | (n = 19) pregnant adolescents (n = 4), adolescent mothers (n = 3), male partners (n = 2), caregivers (n = 2), youth leaders (n = 2), mental health researchers (n = 4), theory of change consultant (n = 1), design thinking consultant (n = 1) |
| Workshop3: August 2021 | Ranking of problems, what worsens the problem and what makes the problem easier | (n = 18) pregnant adolescents (n = 2), adolescent mothers (n = 5), male partners (n = 2), caregivers (n = 2), youth leaders (n = 2), mental health researchers (n = 4), design thinking consultant (n = 1) |
| Workshop4: September 2021 | Visualization and imaginations of ideal situation | (n = 16) pregnant adolescents (n = 4), adolescent mothers (n = 2), male partners (n = 2), caregivers (n = 2), youth leaders (n = 2), mental health researchers (n = 3), design thinking consultant (n = 1) |

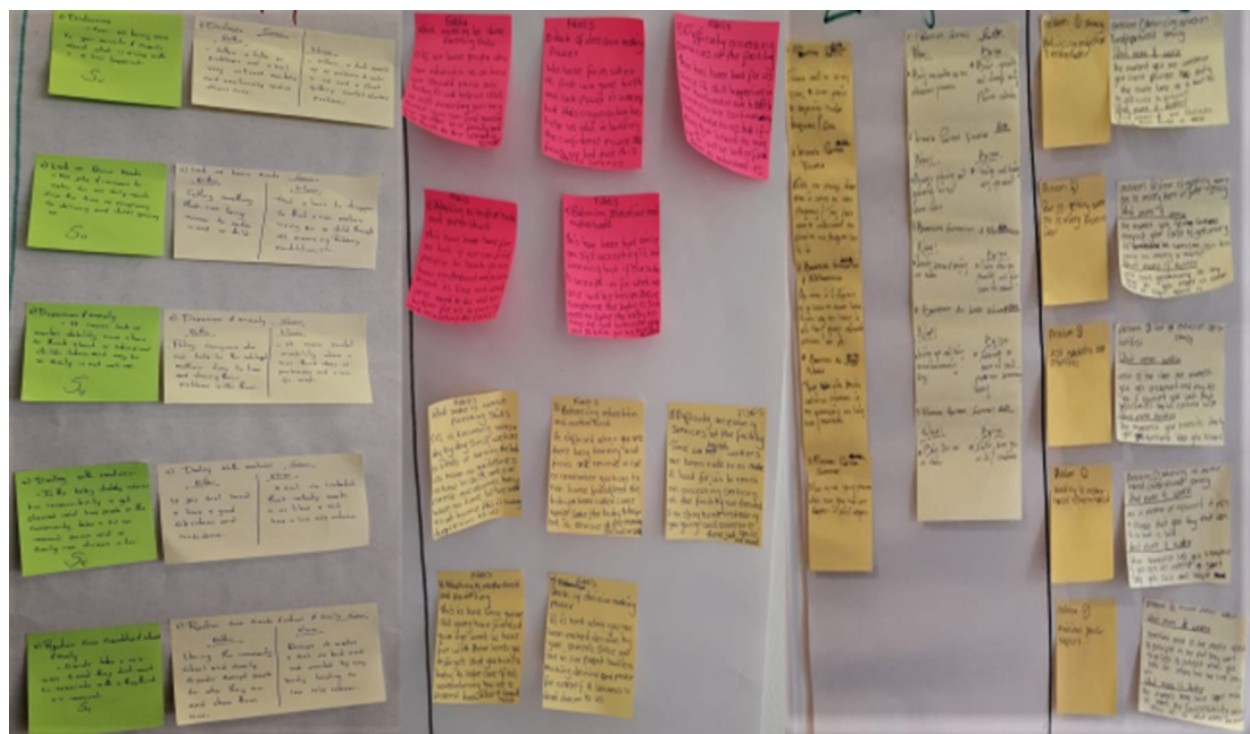

**Fig 2. Workshop process–understanding needs and solutions.**

**Phase 2: Key informant interviews.** We sought written consent to participate after explaining study purpose and objectives, permission was also sought from participants to record the interviews and collected brief sociodemographic data (Table 4). VN, GO, JK and MW conducted the interviews. Interviews were conducted in a large hall that allowed physical distance and, in a tent, whenever the hall was unavailable within the facilities. Whenever these spaces were occupied we met our participants within free stations and rooms assigned to us for these interviews. There is no designated clinic or room for mental health but the facilities share space to run different clinics and programs. We had physical and auditory privacy to the best possible extent given that these are busy primary care facilities. We normally encourage meeting the adolescent mothers with their babies as without them the mothers can become anxious and restless. Our research team that has a number of community health volunteers who would help look after the baby as the mother is interviewed. The language used during interviews was Kiswahili, and each interview lasted for about 30 minutes on average. Audio recordings followed all protocols to ensure confidentiality and data protection including storage in password-protected computer for additional privacy and confidentiality. VN de-identified and transcribed the recordings verbatim. Content was uploaded to NVivo version 10 Qualitative Data Analysis software [20] and thematic content analysis was conducted, identifying emerging themes both deductively and inductively, which were triangulated with the study team's perspectives of the interview transcripts.

## 2.6. Ethical clearance

The study was approved by Kenyatta National Hospital/University of Nairobi ethical review committee (approval no. P694/09/2018). Approval was received from Nairobi County Health no. CMO/NRB/OPR/VOL1/2019/04 and subsequently permit from Kenyan National

Commission for Science, Technology and Innovation (NACOSTI/P/19/77705/28063) was obtained.

## 3. Results

### 3.1. Phase 1

**3.1.1. Participant characteristics.** Twenty-two individuals were recruited to participate in the adolescent advisory and design thinking workshops, including 5 pregnant adolescent girls, and 3 postpartum adolescent girls (Table 3). All 5 of the pregnant adolescents gave birth during the 6-month workshop series.

**3.1.2. Workshop findings.** *Identifying a problem statement.* The first step of the workshops focused on examining key issues and gaining consensus on a problem statement as the foundational component of the design process. Through a series of FGDs, role-plays, and journal entries, three broad problem categories were identified that focused on the adolescent's social ecology, education, and services.

*Social ecology* problems focused on building the skills of caregivers to better support the adolescent including with disclosure and coping, and to reduce blaming from fathers. The need to involve the adolescents' partners was also an identified need so that they can learn to better support pregnant adolescents.

*Education* was another common theme wherein it was agreed that return to school is essential to parenting adolescents' futures. Teachers need to be empowered to support pregnant and parenting adolescents during this critical period in their lives was also one of the workshop findings.

*Services* included building youth responsive services that promote the mental health of pregnant and parenting adolescents, opportunities for life skills training for income generation, and establishing community resource centers embedded within communities that offer recreational and health/sociocultural promotion activities with space allocated to youth. Participants also noted that building communities of practice is an opportunity to form youth groups that focus on mental health promotion. Adolescent participants reported significant gaps in the availability of mental health screening and services; lack of clarity as to which cadres are available to fill mental health service gaps; and the need to create a role within social and welfare systems for mental health awareness and psychological first aid.

**Table 3. Participant sociodemographic description.**

| UCD workshop with HAT advisory | |
| --- | --- |
| **Participants** | **N & mean age** |
| Pregnant adolescents* | 5 (17.4 Years) |
| Adolescent mothers | 3 (18.3 Years) |
| Caregivers | 2 (38 Years) |
| Adolescent fathers | 3 (20.3 Years) |
| Youth leaders | 2 (28 Years) |
| **Research team** | |
| Mental health researchers | 3 |
| Health workers | 2 |
| Design thinking consultant | 1 |
| Theory of change consultant | 1 |

*Gave birth during the course of this study

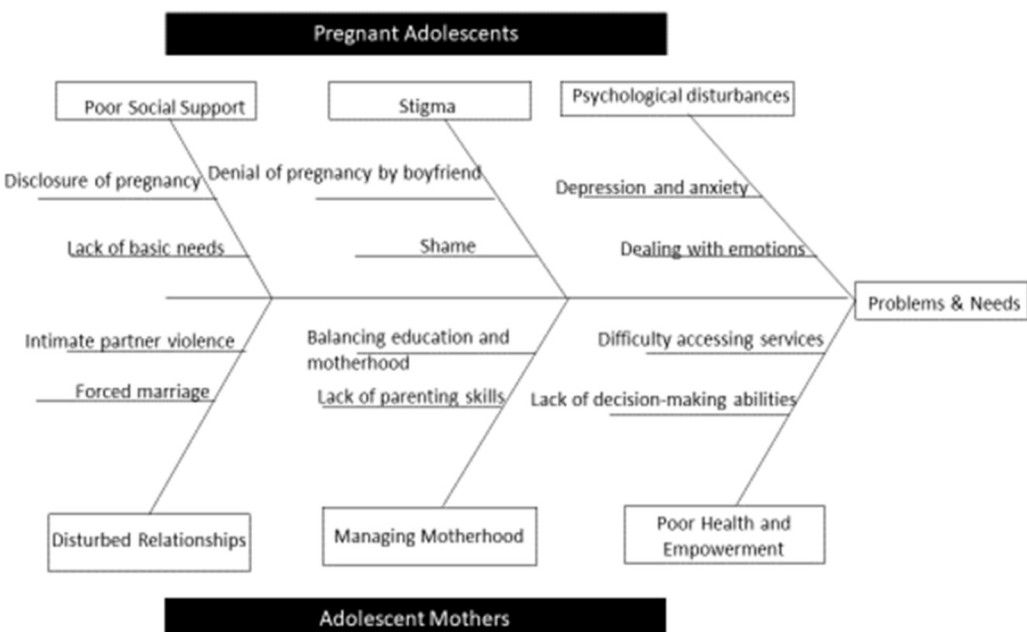

**Fig 3. Problems and needs further deepened through key informant interviews.**

Problem exploration through workshops with adolescents and iterative discussions among mental health researchers and a design thinking specialist led to formulation of a series of problem statements that focused on increasing availability and accessibility of mental health services including improving the quality of adolescent-friendly services that are responsive to the comprehensive needs of adolescent girls and young women as they move through the pregnancy and parenting process. The discussions culminated into a final problem statement: "*how might mental health services be designed to deliver the best possible service, at minimal cost and greater accessibility to pregnant teens and teen moms through the antenatal care (ANC) and postnatal care (PNC) stages?*"

*Examination of the problem and identifying solutions.* Among pregnant adolescents, three common problem themes emerged including peer support, stigma, and psychological disturbances. Within peer support, disclosure of pregnancy and lack of basic needs emerged as common issues. Within stigma, denial of the pregnancy by the boyfriend and pervasive feelings of shame emerged. Within psychological disturbances, common experiences of depression and anxiety and the need to deal with one's emotions emerged.

Among parenting adolescents, three common themes also emerged from Phase 1 work including unstable relationships, managing motherhood, and poor health and empowerment. Within disturbed relationships, forced marriage and interpersonal violence were of primary concern. Within managing motherhood, lack of parenting skills and balancing motherhood and education were significant issues. Within poor health and empowerment, lack of decision-making abilities, and difficulty accessing services were significant challenges (see Fig 3 below).

## 3.2 Phase 2

**3.2.1 Participant characteristics.** Among the 39 key informant interviewees, there were 17 pregnant adolescent girls, and 22 postpartum adolescent girls (Table 4) A WhatsApp-based group was created for adolescent participants for communication purposes which also

**Table 4. Key informant interview participants.**

|  | Categories | Sample (N) | Percentage (%) |
|---|---|---|---|
| Age | 13–15 years | 6 | 15.4 |
|  | 16–18 years | 33 | 84.6 |
| Marital status | Single | 35 | 89.7 |
|  | Married | 4 | 10.3 |
| Education level | Primary | 10 | 25.6 |
|  | Secondary | 29 | 74.4 |
| Gestation | Pregnant | 17 | 43.6 |
|  | Postnatal | 22 | 56.4 |

established an informal network of support and care during the challenging time of the COVID-19 pandemic.

**3.2.2 Key informant interview findings.** Four common themes emerged from key informant interviews (KIIs) including stressors, psychological disturbances, coping with stress, and support (Table 4 presents participant information, see S1 Table and Main Table 5 key themes from the KIIs).

*Stressors.* Economic challenges were a pervasive issue raised by most key informants. The inability of late-term pregnant adolescent girls and young women to work, and lack of available work for pregnant and parenting adolescents and/or their caregivers was a frequent occurrence resulting in high levels of stress. Key informants frequently referred to the concern that they did not have enough to eat, they did not have enough money to purchase nappies for their baby, and that they had insufficient funds to bring their baby to the clinic should the baby become ill. Key informants also noted that they were aware of the increased economic hardship for their caregivers who were under or unemployed and could not afford to help purchase supplies for the baby. Male partners sometimes denied their paternity, disappeared, or refused to assist financially, sometimes due to unemployment, which also contributed to economic hardship.

Fear of delivery was also common. Key informants sometimes voiced insufficient knowledge resulting in fears associated with the laboring process and the body's ability to deliver a baby in a natural way, concerns about the possibility of a cesarean, and for a safe delivery resulting in a healthy baby.

Strained relationships, rejection and experienced stigma were also pervasive issues wherein it became difficult for pregnant and parenting adolescents to maintain friendships due to the inability of friends to understand their new responsibilities. Key informants described multiple instances of verbal harassment from siblings or parents due to their pregnancy and three individuals were forced to leave their homes. Stigma resulted in use of strong, derogatory language towards pregnant and parenting adolescents from neighbors, family members and other community members, and contributed to expulsion from school, and rejection by the male partner including denial of paternity.

*Psychological disturbances.* Feelings of stress and worthlessness due to a lack of anyone to confide in and not feeling loved, resulted in social withdrawal to avoid interactions, and further to feelings of internalized stigma and shame. Suicidal ideation was common with 25.6% (10/39) of key informants reporting thoughts of suicide at some point during the antenatal period. There were no reports of suicidal ideation during the postnatal period. Several key informants discussed their baby as a source of hope to help them move forward. Three key informants reported experiencing sexual violence, and two became pregnant as a result, leading to heightened stress due to associated trauma, in addition to feelings of shame and lack of financial support from the perpetrators.

**Table 5. Critical problems, challenges and needs of our adolescent participants.**

| Theme | Sub theme | Description | Corresponding Vignettes |
|---|---|---|---|
| **Stressors** | Economic challenges | • Poverty; inability to adequately earn livelihood during perinatal period<br>• Inability to provide basic needs for the baby; food, pampers, health care | "Sometimes I want to buy the child's needs, yet I don't see where money will come from"—Adolescent mom<br>"Sometimes we don't eat we sleep hungry"—Adolescent mom |
| | Fear of delivery process | • Apprehensive about the pregnancy outcome, especially the labor process | "My child wants pampers or food, but you know I still cannot get it"–Adolescent mom<br>"I imagine giving birth the way it is going to be painful"—Pregnant adolescent<br>"I am worried because sometimes I can go and be told that my passage is small and they tell me to undergo caesarian section"–Pregnant adolescent |
| | Strained relationships | • Difficulties in relating with family members and friends | "Now that I have a child sometimes my father never comes back to the house at night, I want such things to change"–Adolescent mom<br>"I left my friends, I just have one friend"–Adolescent mom<br>"There is nowhere I can go, I just stay there, but (mother) she can utter painful words but I just stay there"- Adolescent mom |
| | Rejection | • Pregnant adolescent chased away by parent, ignored by friends, suspended from school, rejected by the partner/father of the unborn child. | "You know when I was pregnant, I was sent away from home"–Adolescent mom<br>"I told him that I am pregnant then he cut the communication"-Adolescent mom<br>"I did not think that I can be sent a way in such a style, but they sent me away and told me to get out; they told me to take back the desk, their textbooks and go back home until I deliver that is when they will think about it"–Pregnant adolescent |
| | Stigma | • Use of derogatory language, contempt and despising the pregnant adolescent | "There is a time we were walking then you hear them shouting, 'this thing is going to be a mother'"–Pregnant adolescent<br>". . . despising me and sometimes showing me contempt, they just tell someone, "don't be like CN, you will get pregnant like her"-Pregnant adolescent<br>"When you walk everyone looks at you badly, people talk about you badly and that's why I feel bad having it"–Pregnant adolescent |
| **Psychological disturbance** | Feeling stressed up | • Thinking a lot, wondering whom to confide to. | "I was stressed wondering who will I go to talk to"–Pregnant adolescent<br>"Sometimes I used to feel like I will become insane, I was thinking a lot until I have headache and when I go to hospital, I am told to stop thinking a lot"–Pregnant adolescent<br>"I was feeling stressed"–Adolescent mom |
| | Feeling worthless | • Sensing no love from family and friends, no longer valued | "Sometimes I feel that I am not worthy, I feel that people won't love me because I got pregnant while still young, I feel that my friends will abandon"–Pregnant adolescent<br>"They see me as, like they take me as a person who is not worthy"–Pregnant adolescent<br>"Things that anger me; sometimes my mother talks badly to me, she compares me with my other sister, I feel bad. . . she tells me, "it is better that one, you are not worthy," I feel bad, I even feel like eloping"–Adolescent mom |
| | Withdrawn | • Doesn't want to be around people, distanced from friends, hiding, staying indoors | "Like for me most of the times I feel like there is a time I don't want to be around people, there are times when I just sit be alone" -Pregnant adolescent<br>"When I got pregnant, I got bored of the activities and started disliking them, even at school whenever I am called somehow at least I pretend to be sick so that I don't go"–Pregnant adolescent<br>"I was hiding so that people cannot see me to the extent of not walking"–Adolescent mom |

(Continued)

**Table 5.** (Continued)

| Theme | Sub theme | Description | *Corresponding Vignettes* |
|---|---|---|---|
| | Suicidal | • Thinking of killing oneself/having suicidal thoughts | *"Sometimes I can sit and think of just taking my life"–Pregnant adolescent*<br>*"Sometimes I just feel like killing myself"–Adolescent mom*<br>*"When I was pregnant that is when I had thought about killing myself"–Adolescent mom* |
| **Coping with stress** | Engaging in activities (domestic duties and leisure) | • House chores, watching television, listening to music, walking | *"I just keep myself busy with house chores at least I will not think a lot, but when I am just idle sitting this way, I can even get insane, or on a day that I have not gone to school"–Pregnant adolescent* |
| | Pursuing a hobby | • Photography | |
| | Active social networking | • Visiting friends | *"When I am stressed, I take my elder brother's books and write notes for him to see if the stress will go away or I watch, I switch on the television and watch"–Pregnant adolescent*<br>*"Maybe I just sit or walk around the residential area, doing exercise."–Pregnant adolescent*<br>*"I like listening to songs like hip hops, cooking, cleanliness"–Adolescent mom*<br>*"I just listen to songs"–Adolescent mom*<br>*"Sometimes when I used to go for photography I used to feel as if I am relaxing and fine"–Adolescent mom*<br>*"Maybe I visit my friend and we go for a walk"–Pregnant adolescent* |
| **Support** | Peer support | • Peers are a source of advice, they tell stories, laugh, and feel relief<br>• Especially peers with similar experience | *"The friends I would want are those whom I can tell my problems, we advise each other, and she also tells me, as in we share then help each other"–Adolescent mom*<br>*"My best friend is the one who has helped me and has held my hand"–Pregnant adolescent*<br>*"By the way she (my friend) used to give me hope and telling me, 'I also passed through it, just take heart this life is about perseverance"–Pregnant adolescent* |
| | Family support | • Mothers are more supportive to the adolescent pregnant daughter | *"My mother is the one who gives me hope when she advices me on how life is and how her life used to be, so I normally learn from her nowadays"–Adolescent mom*<br>*"The advice my mother gives me is what gives me hope; she tells me, "slipping is not a fall, you can still continue with your education"–Adolescent mom*<br>*"My mother is the one who has given me hope and being talked to by people"–Pregnant adolescent* |
| | Spiritual support system | • Belief in a supernatural being is a source of hope and courage (Belief that all will be well according to "God's plan"–which includes the pregnancy) | *"You know when God gives you something, he knows that he has given you what you can manage or what you cannot manage, so for me my faith is always in God"–Adolescent mom*<br>*"I said 'I cannot abort because this is something God has given me, and if I tell the person responsible then he denies then it's fine, I am sorry I will just tell my mother, and if my mother also doesn't want then God knows, his plan is fine"–Pregnant adolescent* |
| | Integration back to school | • Adolescent mothers to be supported in resuming school after delivery. | *"Right now, what I can wish is for my child to start walking, then I go back to school"–Adolescent mom*<br>*"I just want after I have given birth, I would tell my mother to take me to a boarding school for girls"–Pregnant adolescent*<br>*"I like reading, I have hope and if I could get someone to help paying school fees, I have hopes of studying; I cannot lose hope in studies"–Adolescent mom* |
| | Livelihood support | • Support in business startup or finding a job for those who are unable to continue with education<br>• Extend support to the mother of the pregnant adolescent (mostly she is the caregiver) | *"Setting up a business for me to do, as long as I build a good future for my child and I that is what I normally think of"–Pregnant adolescent*<br>*"First of all, what I normally wish for is to get job so I can be able to at least support my child"–Adolescent mom*<br>*"Maybe helping my mother"–Pregnant adolescent* |

*(Continued)*

**Table 5.** (Continued)

| Theme | Sub theme | Description | Corresponding Vignettes |
|---|---|---|---|
| | Support groups | • WhatsApp groups appropriate but the challenge is lack of smart mobile phone devices. | *"I would want the support groups"–Adolescent mom*<br>*"Yes, I can be happy (to be in a WhatsApp support group), but the challenge is a phone"–Adolescent mom*<br>*"(The way to get assisted) Mostly the meetings and support group, because for the phone not all people have it"–Adolescent mom* |
| | Mentorship | • Have a guide or mentors to provide appropriate advice. | *"What can help us is having a mentor, just a good mentor"–Adolescent mom*<br>*"The option I would want is maybe somebody who is there to encourage me and give me hope"–Pregnant adolescent*<br>*"At least you offer (adolescent girl) advice so at least she knows 'if I do this and that then this can happen,' then you know also the experiences that one has when pregnant, so when you tell her she gets aware so that at least when she conceives, she knows and understand herself . . .. I would just want at least to be advised"–Pregnant adolescent* |

*Coping with stress.* Key informants often discussed leisure activities at home that helped them cope with stress including reading, watching television, sleeping, house chores, and listening to music. Key informants also described socializing with friends, dancing and photography as helpful to reduce stress.

*Support.* Key informants identified numerous forms of support with the potential to promote mental health during pregnancy. Most often mentioned to support mental health were individuals and entities within the adolescent's social ecology including the value of peers, family, and spiritual support. Mothers of pregnant and parenting adolescents were the family member most often mentioned. Their support, or lack thereof, had a critical influence on the key informants' emotional experiences during pregnancy and postpartum. Financially supportive partners also helped to alleviate stress. Additional sources of support included livelihoods support to reduce economic hardship and facilitate purchasing food and basic items for the baby as well as to return to school. Also mentioned were support groups and mentorships as having the potential extend support and knowledge to less experienced adolescents.

## 4. Discussion

### 4.1. Relevance of this research

A World Health Organization, UNICEF Lancet Commission in 2020 on *the Future of the World's Children* highlighted that investing in the health, education, and development of children not only has positive impacts on the child, but also has positive intergenerational and societal impacts. Strategic investments to address the mental health needs of pregnant and parenting adolescents has the potential to improve health, educational, psychosocial, and developmental outcomes for both mother and baby, while benefiting society at large [21]. Such interventions must be designed in partnership with pregnant and parenting adolescents to identify challenges and co-design and curate solutions.

A key recommendation within the HAT Guidelines on Mental Health Promotive and Preventive Interventions for Adolescents focuses on consideration of psychosocial interventions for pregnant and postpartum adolescents; prioritizing mental health promotion and to improve school attendance. However, while the best evidence came from programs that integrated cognitive-behavioral skills, the evidence is not limited to these programs specifically. The guidelines acknowledge that consideration for integrating psychological interventions

into existing maternal health programs for adolescents is an area of potential importance [15, 22]. Building on the HAT guidance, this study further identifies challenges and recommendations for mental health prevention and promotion. Such exploration from the adolescent perspective provides a detailed understanding that can help to inform and enrich future guidance for pregnant and parenting adolescents.

## 4.2. Mental health prevention

Most adolescents raised the issue of poverty as a significant contributor to poor mental health, highlighting their inability to work during the perinatal period, or to provide basic needs for themselves and their baby including food, diapers, and access to healthcare, including when the baby is ill. Caregivers were often unable to provide additional financial support to assist in purchasing food, infant supplies, or health services. Previous studies in Kenya report similar findings linking poverty to feelings of neglect, depression and suicidality [23, 24]. Studies on pregnant women and poverty have identified direct linkages between food insecurity and feelings of depression [25, 26]. Developing national guidelines and standards that prioritize access to supplementary feeding and other nutrition programs for pregnant and parenting adolescents, as well as including child support grants for young mothers within national budgets and policies may help to prevent some of the poverty related challenges that contribute to poor mental health [27, 28]. Our overall program has involved participation of policy makers and program officials working on adolescent health, mental health and social policy; we hope these social determinants of poor mental health will become areas of direct policy intervention.

To help alleviate poverty, other studies in Kenya and in the region have noted the desire of pregnant and parenting adolescent girls and young women to earn money to cover basic household needs and to reenroll and attend school [29, 30]. Studies that examine interventions that reduce the likelihood of first-time or repeat pregnancy have found that skills training, job placement assistance, vocational training, unconditional household cash transfers, conditional cash transfers based upon school attendance, and purchase of school uniforms can prevent poor mental health [31–35]. Another livelihood and skills training program with support for transition to employment for adolescent girls and young women including pregnant adolescents, found that program participants experienced significantly reduced anxiety around future planning [36].

Reduced access to education was also widely noted as a source of poor mental health among participants. There were accounts of school expulsion due to pregnancy and the inability to pay for school fees, which the participants perceived to significantly limit their life opportunities. While pregnant girls have the right to attend school in Kenya, they are not allowed to return to school until 6 months postpartum and the school re-entry policy does not include provisions to support them to catch up with their classmates upon return [37]. Other studies in Kenya and the region have highlighted the need to address contextual factors including stigma, mistreatment, and financial barriers and to strengthen implementation, monitoring and evaluation of policies for school attendance to optimize school retention and re-entry and to reassure adolescent girls that they have the right to an education regardless of pregnancy or parenting status [37, 38]. Supportive school cultures that have open communication between pregnant and parenting adolescents, caregivers, and teachers and counsellors is critical to prevent further mental stress, and to address their diverse and unique needs including counsellors assisting with pregnancy disclosure support to parents [39], allowing adolescent mothers time to catch up academically with their classmates, and identifying methods to support childcare and breastfeeding on site.

### 4.3. Mental health promotion

In this study, there were numerous instances of participants reporting feeling shunned by friends and family, ostracized, rejected and ignored by partners and family members, and instances of derogatory language resulting in participants socially withdrawing and isolating themselves. Other studies among pregnant adolescents have demonstrated a close correlation between suicidality and poverty, family rejection, social ostracization, and stigma; all commonly experienced by participants within this study [24]. Suicidal ideation during the antenatal period was high (25.6%) among key informants in this study indicating a critical need to ensure that clear standard operating procedures are in place to identify and address suicidal ideation to ensure that pregnant and parenting adolescents are safe and receive mental health support.

Participants mentioned the potential of support groups and mentors to promote mental wellness, prevent feelings of isolation, and to share and learn from other's experiences. A feasibility study in Zimbabwe that examined the use of peer support groups to address feelings of social isolation and stigma among adolescent mothers found that adolescents perceived community-based support groups to have the potential to improve mental health, social support, knowledge sharing, and skills building [40]. Other studies have linked feelings of loneliness associated with adolescent pregnancy to anxiety which has been alleviated by the use of mentors who offer psychosocial support [30, 41]. In Zimbabwe, WhatsApp groups provided additional check-ins between support group meetings [40]. Similarly, in our work, WhatsApp-based group messaging created an environment of support and care aside from strengthening communication demonstrating the potential for a low intensity intervention to promote mental health.

Participants reported instances of poor treatment from providers and stigma leading to further stress and reticence to continue care during antenatal and postnatal appointments. Qualitative studies among pregnant and parenting adolescents across the region have identified poor treatment and stigmatizing attitudes and behaviors from providers as a common cause of disengagement from formal healthcare [42–44]. Qualitative interviews with health providers similarly raise concerns regarding poor treatment towards pregnant and parenting adolescents by other providers, over-medicalization of services, and challenges involving the male partner [44]. Opportunities to address these challenges noted by participants' included provision of youth responsive services that promote the mental health of pregnant and parenting adolescents, which may include mental health screening and service integration, and inclusion of fathers and caregivers in appointments to minimize blame and build psychosocial support.

In response to the common stressors including loss of friendship, social ostracization and feelings of shame when outside of the house, participants identified a variety of mental health promotion opportunities. The importance of spiritual support was a common theme wherein participants noted that spirituality gives them a sense of hope and courage. Similar findings in South Africa noted that even though pregnant and parenting adolescents often were not able to continue attending church later in their pregnancy due to travel, or feelings of stigma, maintaining their faith in a higher power was important to reduce stress. In the same study, some participants noted that older women in the church also would sometimes visit their homes to provide emotional support and mentoring [45]. Another faith-based intervention in Uganda includes taking in pregnant adolescents who have been rejected by their families and partners to provide comprehensive and person-centered care including addressing physical and mental health with a particular focus on empowerment and eventual transition back into the community and family [46]. Building formal linkages and developing programs within faith-based organizations, churches, mosques, and others to support the mental health of pregnant and

parenting adolescent girls and young women is a potentially important area to explore, particularly for those among whom spirituality is important [45].

Study participants commonly experienced strained relationships, rejection, and expulsion from their homes. Mothers and partners were critical determinants of the emotional experiences that adolescent girls and young women had throughout their pregnancies and postpartum. Other studies from Kenya and the region have found mothers of adolescents are overburdened financially and emotionally, but they and other female family members are foundational to the emotional wellbeing of pregnant and parenting adolescents, particularly when the partner is absent [13, 45]. Involving fathers in the pregnancy and parenting process including through encouraging antenatal and postnatal care attendance so that they have a greater understanding of their partner's experiences is also a critical intervention to provide emotional support for their partners and to promote their mental health.

### 4.4. Limitations

While this study provides an important perspective on the mental health challenges of pregnant and parenting adolescents, and prevention and promotion opportunities, there are some limitations. This study did not include the perspectives of healthcare workers, as we wanted to draw out information from the adolescents themselves, who have not yet been studied as extensively through a HCD approach. We explicitly drew out perspectives of adolescents with the eventual aim to sensitize policymakers to those findings. Results from key informant interviews with policy makers have been detailed in a separate manuscript [19]. Given our sample selection process of using an adolescent advisory board to identify eligible participants, the perspectives presented within this paper may be different than had we used an alternative selection process. It is also important to note that the perspectives of key informants are limited to two different health centers under Nairobi Metropolitan services. The perspectives and emotional experiences of pregnant and parenting adolescents in Kenya, and elsewhere in the region may vary.

### 5. Conclusions

To our knowledge, this is the first study of its kind to use a HCD process to draw out the mental health perspectives and experiences of pregnant and parenting adolescent girls and young women. We strongly believe that a HCD approach and taking into account the perspectives of adolescents themselves is essential to designing services that best suits their expressed needs. To address the intersecting risks of mother and baby and promote optimal long-term health outcomes, the challenges and opportunities for mental health prevention and promotion should be integrated into policies and programs to support the mental health needs of pregnant and parenting adolescents in facilities, communities, and their homes. This study revealed that policies that help to alleviate poverty concerns, and programs that offer livelihoods and job placement, and provide cash transfers for school attendance may help to alleviate mental health challenges. Institutions that interact with pregnant and parenting adolescents including health facilities, community organizations and schools should offer adolescent-friendly supportive environments with clear standard operating procedures to rapidly identify and address suicidal ideation. On an interpersonal level, mentoring, supportive supervision and involving fathers also emerged as preferences by study participants.

### Supporting information

**S1 Table. Key informant interview guide.**
(DOCX)

## Acknowledgments

### Availability of data and materials

The datasets used and/or analyzed during the current study available from the corresponding author on reasonable request.

## Author Contributions

**Conceptualization:** Joanna Lai, Manasi Kumar.

**Data curation:** Joseph Kathono, Shillah Mwaniga, Georgina Obonyo, Obadia Yator, Maryann Wambugu, Manasi Kumar.

**Formal analysis:** Georgina Obonyo, Joy Banerjee, Manasi Kumar.

**Funding acquisition:** Joanna Lai, Marcy Levy, Manasi Kumar.

**Investigation:** Joseph Kathono, Shillah Mwaniga, Georgina Obonyo, Joy Banerjee, Joanna Lai, Marcy Levy.

**Methodology:** Joseph Kathono, Vincent Nyongesa, Joy Banerjee, Erica Breuer, Joanna Lai, Manasi Kumar.

**Project administration:** Vincent Nyongesa, Joanna Lai, Marcy Levy.

**Resources:** Joanna Lai.

**Supervision:** Joseph Kathono, Vincent Nyongesa, Shillah Mwaniga, Georgina Obonyo, Obadia Yator, Maryann Wambugu, Joy Banerjee, Marcy Levy.

**Validation:** Joseph Kathono, Vincent Nyongesa, Shillah Mwaniga, Georgina Obonyo, Obadia Yator, Maryann Wambugu, Erica Breuer, Malia Duffy, Joanna Lai, Simon Njuguna.

**Visualization:** Joseph Kathono, Vincent Nyongesa, Shillah Mwaniga, Obadia Yator, Maryann Wambugu, Joy Banerjee, Erica Breuer.

**Writing – original draft:** Manasi Kumar.

**Writing – review & editing:** Vincent Nyongesa, Erica Breuer, Malia Duffy, Simon Njuguna, Manasi Kumar.

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
