## [Decision Letter · Decision Letter 0]

5 Nov 2022

PONE-D-22-21939

Adolescent perspectives on peripartum mental health prevention and promotion from Kenya: Findings from a design thinking approach

PLOS ONE

Dear Dr. Kumar,

Thank you for submitting your manuscript to PLOS ONE. After careful consideration, we feel that it has merit but does not fully meet PLOS ONE’s publication criteria as it currently stands. Therefore, we invite you to submit a revised version of the manuscript that addresses the points raised during the review process.

We look forward to receiving your revised manuscript.

Kind regards,

Frank T. Spradley

Academic Editor

PLOS ONE

7. Please ensure that you refer to Figure 1 and 2 in your text as, if accepted, production will need this reference to link the reader to the figure.

8. We note you have included a table to which you do not refer in the text of your manuscript. Please ensure that you refer to Tables 2 and 4 in your text; if accepted, production will need this reference to link the reader to the Table.

Reviewers' comments:

Reviewer's Responses to Questions

**Comments to the Author**

1. Is the manuscript technically sound, and do the data support the conclusions?

Reviewer #1: Yes

2. Has the statistical analysis been performed appropriately and rigorously? 

Reviewer #1: N/A

3. Have the authors made all data underlying the findings in their manuscript fully available?

Reviewer #1: Yes

4. Is the manuscript presented in an intelligible fashion and written in standard English?

Reviewer #1: Yes

5. Review Comments to the Author

Reviewer #1: Manuscript Number: PONE-D-22-21939

Title: Adolescent perspectives on peripartum mental health prevention and promotion from

Kenya: Findings from a design thinking approach

General comment

The paper addresses an important topic because adolescent mental health vulnerabilities during pregnancy and the postpartum period are critical. Because it focuses on a specific population of adolescents in Kenya, this manuscript makes an important contribution to the literature. Below are some suggestions for improving the manuscript's quality.

Abstract and title

• Title and abstract cover the content of the work.

• Authors can remove references from the abstract part

• It is also preferable if the abstract sections written in independent paragraphs

Introduction

• The introduction is well written. However, the following suggestions will improve the paper

• The authors should clearly state the research gap that they presume to fill. A more explicit description in a few lines would definitely help.

• The objectives described in the manuscript look like those of the main project other than specific to this study. It would be better to clearly state the objectives of this study.

Method

• In general the method part is well written and detail

Analysis: I am not sure on the robustness of the analysis method

Result: Generally good

Discussion looks good

Conclusion: Good

Ethical concern: There are no ethical concerns for this paper

Competing interest: I have no competing interest

6. PLOS authors have the option to publish the peer review history of their article (what does this mean?). If published, this will include your full peer review and any attached files.

Reviewer #1: No

---

## [Author Response · Author response to Decision Letter 0]

6 Dec 2022

we have edited funding and data availability statements

---

## [Decision Letter · Decision Letter 1]

31 Jan 2023

PONE-D-22-21939R1Adolescent perspectives on peripartum mental health prevention and promotion from Kenya: Findings from a design thinking approachPLOS ONE

Dear Dr. Kumar,

Thank you for submitting your manuscript to PLOS ONE. After careful consideration, we feel that it has merit but does not fully meet PLOS ONE’s publication criteria as it currently stands. Therefore, we invite you to submit a revised version of the manuscript that addresses the points raised during the review process.

We look forward to receiving your revised manuscript.

Kind regards,

Adetayo Olorunlana, Ph.D.

Academic Editor

PLOS ONE

Reviewers' comments:

Reviewer's Responses to Questions

**Comments to the Author**

1. If the authors have adequately addressed your comments raised in a previous round of review and you feel that this manuscript is now acceptable for publication, you may indicate that here to bypass the “Comments to the Author” section, enter your conflict of interest statement in the “Confidential to Editor” section, and submit your "Accept" recommendation.

Reviewer #1: All comments have been addressed

Reviewer #2: (No Response)

Reviewer #3: (No Response)

2. Is the manuscript technically sound, and do the data support the conclusions?

Reviewer #1: Yes

Reviewer #2: (No Response)

Reviewer #3: Partly

3. Has the statistical analysis been performed appropriately and rigorously? 

Reviewer #1: N/A

Reviewer #2: (No Response)

Reviewer #3: N/A

4. Have the authors made all data underlying the findings in their manuscript fully available?

Reviewer #1: Yes

Reviewer #2: (No Response)

Reviewer #3: No

5. Is the manuscript presented in an intelligible fashion and written in standard English?

Reviewer #1: Yes

Reviewer #2: (No Response)

Reviewer #3: Yes

6. Review Comments to the Author

Reviewer #1: (No Response)

Reviewer #2: (No Response)

Reviewer #3: The adolescent focus of this work is a strength, and the multi-stage, multi-method research process is exciting to see being implemented with this important target group. I have made a number of remarks and suggestions for strengthening this paper as there are some gaps in the introduction and methodology that were not identified in earlier review stages. I hope these comments can be integrated to increase the clarity and relevance of the manuscript to a wide readership.

In general, this is a really ambitious manuscript. In a sense the authors are trying to both share descriptive findings, as one would for a qualitative work, and also walk through the design-thinking approach and user-centered design lessons. These could ostensibly be two papers—one where challenges are shared, and another where solutions or support avenues are shared. I understand the desire to combine the work but there is a lot to unpack for your reader, and so trying to organize and streamline this as much as possible would be ideal (some of my comments are suggesting ways to do this).

Additionally, I am struggling with the framing of user-centered design in part because it feels like this is more reflecting a participatory form of research—and it is not clear to me what services, systems, or interventions are being workshopped (aside from maybe the “support” lens of the KIIs). I know these two concepts are not far from one another, however, I would suggest that the authors think critically about how to firstly, define user-centered design, and secondly, make a strong case for what this means in the public health/research space and the specific study context to really “convince” your readers.

Introduction:

1. The framing of the introduction could be improved. The suicide statistic for the general adult population could be placed after the opening sentence of the introduction, where general information about adolescent mental health is being shared. The pivot to “mental health promotion” in the second paragraph is key, but should be framed more around opportunities – right now, this entire paragraph (and especially the second part of the sentence, on suicide) is framed around risks and stressors. Can more be shared about what mental health promotion activities look like? Beyond this, the paper should then narrow to stay focused on adolescent prenatal/postpartum mental health, which is different in scope and the focus of this paper. It generally does this well.

2. The authors of the paper should work to describe the gap they seek to address more concretely and clearly. Right now, it sounds like there is good understanding of challenges surrounding pregnancy, and of the feasibility of integrating mental health and maternal care services. Is the gap that antenatal attendance is low – meaning these services can’t be provided? Is it that we know these are feasible, but have not yet been rolled out? A few clear sentences specifically stating what is known, and where specific gaps exist that you will address, are required.

3. I would suggest the following re-write, for clarity: “This study took place within the contexts of two related studies: 1) the Helping Adolescents Thrive (HAT) Kenya program (jointly delivered by the World Health Organization and UNICEF) and 2) the INSPIRE study (‘Implementing mental health interventions for pregnant adolescents in primary care LMIC settings’). The focus of both studies is on peripartum adolescent women in Kenya.”

4. It would be useful to explain “the HAT framework” before you describe using it to structure you two objectives. What does this framework entail exactly? This can and should be part of the introduction.

5. Young fathers are mentioned in the abstract, but the two objectives are focused on data gathering from pregnant and parenting adolescent women. Can this be explained up front?

6. Given the focus on mental health promotion, is Objective 1 also gathering data on key contributors to mental wellbeing? Or coping strategies to foster wellbeing?

Method:

7. It would be useful to understand how many people were part of the advisory board, and also if the group always met together – e.g. if so, there may be intergenerational or other power dynamics across the types of group members listed that are worth acknowledging.

8. It is not clear what “the adolescent workshops” are referring to in the study design section. Looking ahead, it seems that the information is present later on, but this does not make sense at present. I would suggest explaining, explicitly, the three phases (even in a table?) with the structure/design of each, who was involved in each, and then setting the stage to describe in more detail each of these groups/activities later on. E.g. “The study consisted of three phases: 1) advisory board formation and input, 2) adolescent workshops, and 3) key informant interviews. In Phase 1, xxx. In Phase 2, we engaged xxx. Focus group discussions were employed, alongside role-plays and journals of adolescents’ daily activities. These activities helped xxx. In Phase 3, xxx” This would be much clearer for your reader; the figure is somewhat useful but is lacking in its descriptive content and clarity. I also found it confusing as I kept reading to see the advisory activities merged with the design thinking meetings. I would suggest these be reported separately to help your reader understand better how these differ.

9. I am not sure it is useful to bring in policymaker interviews, as these are not clearly introduced and the data are not shared in this manuscript.

10. For clarity, I might start the Methods section with the setting, to contextualize what the activities and sample look like. The setting should also include relevant information about the population in question, beyond administrative aspects – what is the adolescent pregnancy rate? Why was this area chosen?

11. It is not clear whether the participants in the adolescent workshops were all taken from the advisory board. This will be important to clarify.

12. Are there any bias considerations or concerns with youth advocates selecting advisory board members? As these members had a role in scoping the research, were there any voices that may have been omitted, even unintentionally, and were any efforts made to consider this?

13. The description of key informant interviews should clarify if the reimbursements were in line with ethics requirements/standards. Additionally, clarity is needed about the process of consent/assent for those adolescents under 18 (as age-of-consent laws vary by country context).

14. The workshop dates seem to be out of order. In line with typical reporting, however, I would state that the workshops took place between April-September 2021 and call these Workshops 1, 2, 3, 4 in the table.

15. It would be interesting within the table to see the breakdown of the 28 participants e.g. who are pregnant adolescents? Who are CBO representatives? If there is 1 adolescent mother present, versus 12, this is important context.

16. Information about the KIIs should include language of interview and how long the interviews took, on average.

17. Under data collection and analysis, Phase 1 and 2 should be presented in order. There is still no information about assent or parental consent, which is crucial to include given the vulnerable profile of this study population and topic matter.

18. I would clarify that focus group discussions were interactive and participatory in nature, and during each of these discussions, activities include breakout group meetings and brainstorming sessions. This was not immediately clear as I first read it.

Results

19. The first sentences of the Results section should be capitalized when described participants.

20. Suggested edit: “Male partners sometimes denied their paternity, disappeared, or refused to assist financially, sometimes due to unemployment, which also contributed to economic hardship.”

21. In the table, I would be careful of using the phrasing “supernatural being” but maybe rather “spiritual beliefs as a source of hope xxx” as this is more expansive and accepted terminology.

22. I would suggest reshuffling your Results: Phase 1: participant characteristics, findings and then Phase 2: participant characteristics, findings. It is very fragmented at the moment and makes it harder to follow emerging themes from each.

23. Although the HAT framework is referenced above, it is not clear how this framework has guided analysis or emerging findings. Can this be a more explicit thread throughout the paper?

24. The second objective (identify preferences for mental health promotion activities) does not seem to be explicitly linked here. The data shared, especially from KIIs, feels like it has come directly from the participants – but are these preferences for specific activities, or just ways of coping, support figures, challenges etc currently in use? Is there a workshop or synthesis later on, where these preferences are actually talked about, or divided up from general themes? Right now, the discussion seems to read like a qualitative paper normally would, and while this is fine, I think you should try to make the link to identifying preferences or opportunities to target later on for mental health promotion—or, remove this as an overarching objective from this paper? This is an important conceptual distinction, as right now I don’t believe your second objective is being clearly met in the current presentation of the data.

Discussion

25. As I mentioned above, more discussion of the role of multiple stakeholders (adults and adolescents) in the design workshops could be useful. In some ways, these workshops seem broader and not necessarily as adolescent-user-designed as the paper describes.

26. A note about the HAT guidelines – while the best evidence came from programs that integrated cognitive-behavioral skills, the evidence is not limited to these programs specifically, so it may be worth omitting that part of your statement.

In the conclusion, it would be useful to address the changes I have suggested at the top of these comments, re: clear definition of what user-centered design looks like and how you are operationalizing it, alongside some of the other considerations of scope noted above.

General typos/grammar edits:

- Helping Adolescents Thrive should not have an apostrophe

- I would not use “adolescent women”— I think adolescent girls and young women is probably most appropriate, or pregnant and parenting adolescents more broadly

- Check for appropriate uppercase/lowercase (e.g. “Kariobangi North”, “Kangemi Health Center” in 2.2 Settings, “Phase 1”, “Table 1”, “Supplementary File 1” everywhere these are referenced, etc.)

7. PLOS authors have the option to publish the peer review history of their article (what does this mean?). If published, this will include your full peer review and any attached files.

Reviewer #1: No

Reviewer #2: No

Reviewer #3: **Yes: **Christina A. Laurenzi

---

## [Author Response · Author response to Decision Letter 1]

8 Jun 2023

Dear Editor 

pl find our response to the reviewer comments also attached in the files. 

regards MAnasi 

Response to the reviewer feedback 28th March 2023

PONE-D-22-21939

Adolescent perspectives on peripartum mental health prevention and promotion from Kenya: Findings from a design thinking approach

PLOS ONE

Dear Editor 

We thank you for you feedback and review of the above mentioned paper and please find attached a 

• A rebuttal letter that responds to each point raised by the academic editor and reviewer(s). 

• A marked-up copy of manuscript that highlights changes made to the original version. 

• An unmarked version of revised paper without tracked changes. 

Response to the editor requests (In attachment)

This manuscript reports a gap in adapting interventions for peripartum adolescents. However, the theory of Change workshop i.e. advisory group discussions with FGDs, journals etc should be a combined manuscript to avoid salami slicing. If available, including data from young fathers could help in strengthening the manuscript. The manuscript also seems to address “adolescent perspectives on peripartum mental health prevention and promotion from Kenya” yet a lot of information is provided on the advisory group. Although it was the same study, it is not clear how this information is responding to the proposed AIMs. Otherwise, the title can change to reflect this.

1. Remove references from the abstract.

Response: We thank you for this feedback and have now removed the references 

2. Check for no spacing, double punctuation marks, spaces etc throughout the manuscript

Response: we thank you and have now amended this 

3. While this statement is true” Global prevalence studies demonstrate a wide variation in estimates of postpartum depression among adolescents, ranging between 14-53%”, it is not clear how it is linked to the statement before and after. The introduction needs to flow. 

Response: we have rectified this now. 

4. Page 5: “A recent program evaluation in Kenya”……describes an existing program, yet the statement that follows “In the context of Kenya however, further understanding factors from the…” contradicts this earlier statement and the factors to be explored are not stated. The aim of the proposed study is also not clearly stated.

Response: we have amended this (See page 5)

5. The design thinking process needs to be explained and its rationale discussed in the manuscript – perhaps at the introduction section.

Response: we have explained this now 

Methods:

1. Elaborate on the advisory groups? How was data collected? How many FGDs were conducted? How many participants were in each FGD? Is this what the author’s refer to as Theory of Change workshops? The way the methods section is structured is a bit confusing to the reader.

Response: this is part of the TOC and design thinking process to bring on board the participants and relevant stakeholders 

2. There was an advisory board consisting of various stakeholders yet only discussions with pregnant women and policy makers were conducted and reported separately. Was there a rationale for this? 

Response: we have carried out interviews with health facility workers, did work on priority setting with program managers and facility workers and trainings etc but focus on the two groups was key here. Gaps in understanding of what is promotion and how it is done in policy space

3. 17-22 key informant interviews seem a high number for a homogenous group of pregnant mothers. What was the rationale for this number? 

Response: we did need a range of young people and they were involved in designing solutions 

4. A WhatsApp-based group is only introduced in the results section rather than the methods section.

Response: Thank you for noting this, we have now incorporated that bit in on page 7

Response to the reviewer #3 

Reviewer #3: The adolescent focus of this work is a strength, and the multi-stage, multi-method research process is exciting to see being implemented with this important target group. I have made a number of remarks and suggestions for strengthening this paper as there are some gaps in the introduction and methodology that were not identified in earlier review stages. I hope these comments can be integrated to increase the clarity and relevance of the manuscript to a wide readership. In general, this is a really ambitious manuscript. In a sense the authors are trying to both share descriptive findings, as one would for a qualitative work, and also walk through the design-thinking approach and user-centered design lessons. These could ostensibly be two papers—one where challenges are shared, and another where solutions or support avenues are shared. I understand the desire to combine the work but there is a lot to unpack for your reader, and so trying to organize and streamline this as much as possible would be ideal (some of my comments are suggesting ways to do this). Additionally, I am struggling with the framing of user-centered design in part because it feels like this is more reflecting a participatory form of research—and it is not clear to me what services, systems, or interventions are being workshopped (aside from maybe the “support” lens of the KIIs). I know these two concepts are not far from one another, however, I would suggest that the authors think critically about how to firstly, define user-centered design, and secondly, make a strong case for what this means in the public health/research space and the specific study context to really “convince” your readers.

Response: We really appreciate your feedback on the complexity of the content here. I think you are right that the work is fundamentally about participatory research but it is curating, exploring, planning mental health promotion from the perspective of pregnant and parenting adolescents. It is in this sense that the work adopts a design thinking approach. In human-centered thinking/design thinking the problem is tackled by framing different facets where the challenges are experienced, what emotions, thought processes, socio-familial and cultural processes occur alongside are explored as well. initially the idea was to break it down into two papers but one of the earlier journals where we submitted this paper and a few others gave strong feedback about combining the two and presenting a more ‘complex and interlinked’ picture as she recommended us. We have emphasized that the work is driven by participatory method and tried to highlight what we mean by designing the inquiry process- along with our participants and being led by core rubrics of design thinking. 

Pl also see: https://designthinking.ideo.com/#:~:text=Design%20Thinking%20Defined,the%20requirements%20for%20business%20success.

Introduction:

1. The framing of the introduction could be improved. The suicide statistic for the general adult population could be placed after the opening sentence of the introduction, where general information about adolescent mental health is being shared. The pivot to “mental health promotion” in the second paragraph is key, but should be framed more around opportunities – right now, this entire paragraph (and especially the second part of the sentence, on suicide) is framed around risks and stressors. Can more be shared about what mental health promotion activities look like? Beyond this, the paper should then narrow to stay focused on adolescent prenatal/postpartum mental health, which is different in scope and the focus of this paper. It generally does this well.

Response: Thank you for this comment. We have moved the sentence about suicide to the first paragraph. The third paragraph moves into examples of mental health promotion activities for pregnant adolescents available in the literature. 

2. The authors of the paper should work to describe the gap they seek to address more concretely and clearly. Right now, it sounds like there is good understanding of challenges surrounding pregnancy, and of the feasibility of integrating mental health and maternal care services. Is the gap that antenatal attendance is low – meaning these services can’t be provided? Is it that we know these are feasible, but have not yet been rolled out? A few clear sentences specifically stating what is known, and where specific gaps exist that you will address, are required.

Response: Thank you for this comment. We have changed this sentence to the following: In the context of Kenya however, further understanding factors from the perspective of pregnant and parenting adolescents themselves that help prevent mental stress and promote mental health are critical to optimize antenatal and postnatal care attendance so that it supports the mental health of pregnant and postpartum adolescent girls and improves and infant and maternal health outcomes.

3. I would suggest the following re-write, for clarity: “This study took place within the contexts of two related studies: 1) the Helping Adolescents Thrive (HAT) Kenya program (jointly delivered by the World Health Organization and UNICEF) and 2) the INSPIRE study (‘Implementing mental health interventions for pregnant adolescents in primary care LMIC settings’). The focus of both studies is on peripartum adolescent women in Kenya.”

Response: Thank you. We have changed the sentence accordingly.

4. It would be useful to explain “the HAT framework” before you describe using it to structure you two objectives. What does this framework entail exactly? This can and should be part of the introduction.

Response: HAT framework is published by WHO/UNICEF. pl see page 5 for more information. See the following available guidance: Helping adolescents thrive toolkit: strategies to promote and protect adolescent mental health and reduce self-harm and other risk behaviours. Geneva: World Health Organization and the United Nations Children’s Fund (UNICEF), 2021. Licence: CC BY-NC-SA 3.0 IGO

5. Young fathers are mentioned in the abstract, but the two objectives are focused on data gathering from pregnant and parenting adolescent women. Can this be explained up front?

Response: young fathers are important part of strengthening of care for perinatal adolescents and we have now presented this upfront. We have explained it on pages 3,6 and 9 now. 

6. Given the focus on mental health promotion, is Objective 1 also gathering data on key contributors to mental wellbeing? Or coping strategies to foster wellbeing?

Response: Yes, this research also gathered information on key contributors to mental wellbeing/coping strategies. We have clarified this as such within the manuscript. 

Method:

7. It would be useful to understand how many people were part of the advisory board, and also if the group always met together – e.g. if so, there may be intergenerational or other power dynamics across the types of group members listed that are worth acknowledging.

Response: the advisory group included policy makers, lived experience representation by adult women and adolescent girls as well as a range of caregivers who were nominated by adolescents and several of our county program officers in consultation. There are 4 meetings overall in this 1.5 years study and we did not find dynamics that were oppositional. There were different interests and understanding between stakeholders but lots of goodwill. 

8. It is not clear what “the adolescent workshops” are referring to in the study design section. Looking ahead, it seems that the information is present later on, but this does not make sense at present. I would suggest explaining, explicitly, the three phases (even in a table?) with the structure/design of each, who was involved in each, and then setting the stage to describe in more detail each of these groups/activities later on. E.g. “The study consisted of three phases: 1) advisory board formation and input, 2) adolescent workshops, and 3) key informant interviews. In Phase 1, xxx. In Phase 2, we engaged xxx. Focus group discussions were employed, alongside role-plays and journals of adolescents’ daily activities. These activities helped xxx. In Phase 3, xxx” This would be much clearer for your reader; the figure is somewhat useful but is lacking in its descriptive content and clarity. I also found it confusing as I kept reading to see the advisory activities merged with the design thinking meetings. I would suggest these be reported separately to help your reader understand better how these differ.

Response: Thank you for these helpful comments and suggestions. We have created a box to clarify the two different phases of the research. Design process is not a formal research process but much of the work is iterative and blended with action research. we have now tried to clarify this. 

9. I am not sure it is useful to bring in policymaker interviews, as these are not clearly introduced and the data are not shared in this manuscript.

Response: we thank you for this comment. We do have a manuscript that has been accepted now and we have pre-empted this inquiry in the beginning. 

10. For clarity, I might start the Methods section with the setting, to contextualize what the activities and sample look like. The setting should also include relevant information about the population in question, beyond administrative aspects – what is the adolescent pregnancy rate? Why was this area chosen?

Response: Thank you for this suggestion. We have moved the ‘Setting’ paragraph to the first part of the methods. There were high perinatal adolescent population and visits to MCH in the sites that were first included in the study in 2018 and the HAT study leveraged the two sites where we were carrying out interventions under a grant that is testing group interpersonal psychotherapy for pregnant adolescents. Almost 2-3 adolescents of ages 14-19 per week (8-12 per month and on average 140 plus adolescents a year from these two centers). At the time these centers were chosen both reported extremely high rates of pregnancy in young adolescent girls. 

11. It is not clear whether the participants in the adolescent workshops were all taken from the advisory board. This will be important to clarify.

Response: Yes, these were taken from the advisory board. Two caregivers of adolescents were selected from the community to join the advisory board to bring out their experiences in taking care of pregnant adolescents or adolescent mothers. 

12. Are there any bias considerations or concerns with youth advocates selecting advisory board members? As these members had a role in scoping the research, were there any voices that may have been omitted, even unintentionally, and were any efforts made to consider this?

Response: we don’t think there are biases. Their experience of motherhood, pregnancy and parenthood or being a caregiver were bases for participation. design workshops and theory of change processes are contextual and driven by ground level stakeholders. The intention is not to say that all views represented here sum up the entirety of the problem associated with adolescent parenthood. The intention is to have a balanced representation and uphold perspectives of the participants to share the problems around mental health promotion and prevention as these exist. We have not omitted voices or consciously or unconsciously deleted representation. We are concerned that the reviewer thinks there would be omission. The advisory group members were very synergistic and there was alignment in views however ability to act on these could vary based on constraints of different stakeholders. 

13. The description of key informant interviews should clarify if the reimbursements were in line with ethics requirements/standards. Additionally, clarity is needed about the process of consent/assent for those adolescents under 18 (as age-of-consent laws vary by country context).

Response: Thank you; participant reimbursements are a requirement and align with guidance from Kenyatta National Hospital/University of Nairobi ethical review committee. With regards to consent/assent, in Kenya pregnant adolescents are considered emancipated minors and are able to consent regardless of their age. We have clarified these two on page 9 

14. The workshop dates seem to be out of order. In line with typical reporting, however, I would state that the workshops took place between April-September 2021 and call these Workshops 1, 2, 3, 4 in the table.

Response: Thank you for your suggestion; we have now corrected the dates and incorporated Workshop numbering per your suggestions (See page 9 and 10)

15. It would be interesting within the table to see the breakdown of the 28 participants e.g. who are pregnant adolescents? Who are CBO representatives? If there is 1 adolescent mother present, versus 12, this is important context.

Response: we have provided this information now in table 3. 

16. Information about the KIIs should include language of interview and how long the interviews took, on average.

Response: Thank you; we have included a statement on language of interview and average time (See page 11)

17. Under data collection and analysis, Phase 1 and 2 should be presented in order. There is still no information about assent or parental consent, which is crucial to include given the vulnerable profile of this study population and topic matter.

Response: Thank you for pointing this out. We have placed them in the correct order. We have also included information about assent and consent (see page 9)

18. I would clarify that focus group discussions were interactive and participatory in nature, and during each of these discussions, activities include breakout group meetings and brainstorming sessions. This was not immediately clear as I first read it.

Response: Thank you. We have clarified this within the manuscript. 

Results

19. The first sentences of the Results section should be capitalized when described participants.

Response: Thank you. We have capitalized this.

20. Suggested edit: “Male partners sometimes denied their paternity, disappeared, or refused to assist financially, sometimes due to unemployment, which also contributed to economic hardship.”

Response: Thank you. We have included this suggestion. 

21. In the table, I would be careful of using the phrasing “supernatural being” but maybe rather “spiritual beliefs as a source of hope xxx” as this is more expansive and accepted terminology.

Response: we thank you and have amended this change. This is indeed a more sound language. 

22. I would suggest reshuffling your Results: Phase 1: participant characteristics, findings and then Phase 2: participant characteristics, findings. It is very fragmented at the moment and makes it harder to follow emerging themes from each.

Response: Thank you for these suggestions. We have adapted the manuscript accordingly.

23. Although the HAT framework is referenced above, it is not clear how this framework has guided analysis or emerging findings. Can this be a more explicit thread throughout the paper?

Response: we have now added this information. HAT is evolving as a framework and was recently launched. Our work is a pilot to extend practice implementation as culturally rooted theoretical models of testing it. Pl see our discussion page 32.

24. The second objective (identify preferences for mental health promotion activities) does not seem to be explicitly linked here. The data shared, especially from KIIs, feels like it has come directly from the participants – but are these preferences for specific activities, or just ways of coping, support figures, challenges etc currently in use? Is there a workshop or synthesis later on, where these preferences are actually talked about, or divided up from general themes? Right now, the discussion seems to read like a qualitative paper normally would, and while this is fine, I think you should try to make the link to identifying preferences or opportunities to target later on for mental health promotion—or, remove this as an overarching objective from this paper? This is an important conceptual distinction, as right now I don’t believe your second objective is being clearly met in the current presentation of the data.

Response: we have removed this from the objective as per your suggestion. 

25. As I mentioned above, more discussion of the role of multiple stakeholders (adults and adolescents) in the design workshops could be useful. In some ways, these workshops seem broader and not necessarily as adolescent-user-designed as the paper describes.

Response: the broad remit of the workshop appears so as it includes stakeholders who inform, interface and work closely with adolescent and shape their development, well-being and health behaviors and preferences. In the discussion on page 32 this has been highlighted. 

26. A note about the HAT guidelines – while the best evidence came from programs that integrated cognitive-behavioral skills, the evidence is not limited to these programs specifically, so it may be worth omitting that part of your statement.

Response: we have alludes to HAT strategies and toolkit paper that extends this work but we have reflected on your comments and amended the citation of the work to include the toolkit. 

In the conclusion, it would be useful to address the changes I have suggested at the top of these comments, re: clear definition of what user-centered design looks like and how you are operationalizing it, alongside some of the other considerations of scope noted above.

Response: we thank you for this note and have added some comments. We hope you will find it satisfactory.

General typos/grammar edits:

- Helping Adolescents Thrive should not have an apostrophe 

Response: We removed the apostrophe.

- I would not use “adolescent women”— I think adolescent girls and young women is probably most appropriate, or pregnant and parenting adolescents more broadly.

Response: We have changed to pregnant and parenting adolescent girls and young women. 

- Check for appropriate uppercase/lowercase (e.g. “Kariobangi North”, “Kangemi Health Center” in 2.2 Settings, “Phase 1”, “Table 1”, “Supplementary File 1” everywhere these are referenced, etc.).

Response: Thank you. We have checked them.

---

## [Decision Letter · Decision Letter 2]

19 Jun 2023

PONE-D-22-21939R2Adolescent perspectives on peripartum mental health prevention and promotion from Kenya: Findings from a design thinking approachPLOS ONE

Dear Dr. Kumar,

Thank you for submitting your manuscript to PLOS ONE. After careful consideration, we feel that it has merit but does not fully meet PLOS ONE’s publication criteria as it currently stands. Therefore, we invite you to submit a revised version of the manuscript that addresses the points raised during the review process.

We look forward to receiving your revised manuscript.

Kind regards,

Adetayo Olorunlana, Ph.D.

Academic Editor

PLOS ONE

Journal Requirements:

Reviewers' comments:

Reviewer's Responses to Questions

**Comments to the Author**

1. If the authors have adequately addressed your comments raised in a previous round of review and you feel that this manuscript is now acceptable for publication, you may indicate that here to bypass the “Comments to the Author” section, enter your conflict of interest statement in the “Confidential to Editor” section, and submit your "Accept" recommendation.

Reviewer #3: (No Response)

2. Is the manuscript technically sound, and do the data support the conclusions?

Reviewer #3: Yes

3. Has the statistical analysis been performed appropriately and rigorously? 

Reviewer #3: N/A

4. Have the authors made all data underlying the findings in their manuscript fully available?

Reviewer #3: No

5. Is the manuscript presented in an intelligible fashion and written in standard English?

Reviewer #3: Yes

6. Review Comments to the Author

Reviewer #3: The authors have made significant improvements to this article. I noted a few outstanding areas to be addressed ahead of publication, although the core changes required have been made.

Major comments

1. In response to the earlier note about unintentional exclusion, this should be considered as a comment about how embedded social and behavioral patterns and ways of doing things can and often do play out in the context of a research project. A note about ethical considerations in recruitment would be encouraged, wherever the authors feel it makes sense, given the vulnerability of this population and the topic of focus.

2. For Table 2, it would be useful to have the breakdown of participants by role (e.g. n=28 is the total group, but how many are pregnant adolescents? CBO representatives? Etc). This is standard reporting for multi-stakeholder groups. Ideally, notation would be “Pregnant adolescents (n=xx), adolescent mothers (n=xx), male partners (n=xx)” and so on.

3. It would be ideal to note the specific location of the KIIs, beyond “within health facilities”, as this may not denote a private space. Were there any considerations around privacy? Also, what about childcare—were there any challenges around arranging one-on-one time with peripartum young women?

4. It may be important, along with the ethical considerations add-in suggested above, to note any considerations for engaging young women who had just given birth over the course of the workshop series. Were any special provisions made to support their involvement, or was this not possible following the birth of their infant? This could be a useful lesson for others seeking to replicate some of this work.

5. The headings in Results need to be better differentiated and should be leveled by phase. If you use “3. Results” then 3.1 should be Phase 1, with 3.1.1 denoting participant characteristics e.g. and 3.2 can indicate Phase 2.

6. The purpose of the WhatsApp group for Phase 2 participants is not evident; was this created following recruitment/the interview itself? Was this with fellow young mothers or with researchers too?

7. A previous comment was not integrated into the discussion, and I copy it here: “while the best evidence came from programs that integrated cognitive-behavioral skills, the evidence is not limited to these programs specifically, so it may be worth omitting that part of your statement.” The discussion still reads that “[the evidence] is limited to cognitive behavioural skills-building programs” which is not accurate and should be revised in line with the above comment.

Minor/copy comments

8. A copy edit would be beneficial, as a number of minor typos persist throughout. A few examples: 1) caregiver relationships (not relationship) in the section detailing HAT, 2) if a health center is named, it should be capitalized, e.g. “Kangemi Health Center”, 3) Phase 1 or Phase 2 should always be capitalized

9. For the aim of the study at the introduction’s end, the addition of men should include more specifically “pregnant and parenting adolescent girls and young women as well as adolescent boys and young men”.

10. It would be important to cite the separate manuscript in which interviews with policymakers are documented.

11. It seems as though the same photo has been inserted twice, I am not sure if this is an error on the PDF creation software but just wanted to flag this.

12. Figure 3 is very hard to read and should be made higher-resolution/larger, or attached separately.

13. Table 5 is difficult to read and should be formatted to be left-justified to improve legibility.

14. There is a remaining comment within the discussion of the manuscript that should be deleted before re-submission.

7. PLOS authors have the option to publish the peer review history of their article (what does this mean?). If published, this will include your full peer review and any attached files.

Reviewer #3: No

---

## [Author Response · Author response to Decision Letter 2]

6 Aug 2023

Dear Editor we have added comments and amendments based on reviewer feedback 

regards Manasi

---

## [Editor Report · Decision Letter 3]

18 Aug 2023

Adolescent perspectives on peripartum mental health prevention and promotion from Kenya: Findings from a design thinking approach

PONE-D-22-21939R3

Dear Dr. Kumar,

We’re pleased to inform you that your manuscript has been judged scientifically suitable for publication and will be formally accepted for publication once it meets all outstanding technical requirements.

Kind regards,

Adetayo Olorunlana, Ph.D.

Academic Editor

PLOS ONE

Additional Editor Comments (optional):

If appropriate kindly move the **highlights **after the Keywords to methods session before (2.1). The article will flow better without highlight preceding the introduction. If not appropriate the article loses nothing if authors outrightly delete the highlights. 
---

## [Editor Report · Acceptance letter]

29 Aug 2023

PONE-D-22-21939R3 

Adolescent perspectives on peripartum mental health prevention and promotion from Kenya: Findings from a design thinking approach 

Dear Dr. Kumar:

I'm pleased to inform you that your manuscript has been deemed suitable for publication in PLOS ONE. Congratulations! Your manuscript is now with our production department. 

Kind regards, 

on behalf of

Associate Professor Adetayo Olorunlana 

Academic Editor

PLOS ONE